# Analytical Gate Delay Variation Model with Temperature Effects in Near-Threshold Region Based on Log-Skew-Normal Distribution

**Jingjing Guo** **, Peng Cao, Jiangping Wu, Zhiyuan Liu and Jun Yang** *

National ASIC System Engineering Center, Southeast University, Nanjing 210096, China; guojingjing@seu.edu.cn (J.G.); caopeng@seu.edu.cn (P.C.); wujiangping@seu.edu.cn (J.W.); zyliuasic@seu.edu.cn (Z.L.)
*   Correspondence: dragon@seu.edu.cn; Tel.: +86-025-8379-3265

**Abstract:** The near-threshold design is widely employed in the energy-efficient circuits, but it suffers from a high sensitivity to process variation, which leads to 2X delay variation due to temperature effects. Hence, it is not negligible. In this paper, we propose an analytical model for gate delay variation considering temperature effects in the near-threshold region. The delay variation model is constructed based on the log-skew-normal distribution by moment matching. Moreover, to deal with complex gates, a multi-variate threshold voltage approximation approach of stacked transistors is proposed. Also, three delay metrics (delay variability, $\pm 3\sigma$ percentile points) are quantified and have a comparison with other known works. Experimental results show that the maximum of delay variability is 5% compared with Monte Carlo simulation and improves 5X in stacked gates compared with lognormal distribution. Additionally, it is worth mentioning that, the proposed model exhibits excellent advantages on $-3\sigma$ and stacked gates, which improves 5X–10X in accuracy compared with other works.

**Keywords:** temperature; delay variation; near-threshold; log-skew-normal distribution

## 1. Introduction

By technology downscaling deep into the nanometer era, the near-threshold design has become one of the most efficient ways for high energy efficiency application [1–4]. However, it suffers from a much higher sensitivity to process, voltage and temperature variations [5–7]. In view of transistor operation mechanisms, MOSFET changes from strong inversion to weak inversion with decreasing voltage, and its primary current also changes from drift to diffusion current [8]. In the near-threshold, since these two kinds of current cannot be ignored, its current formula is different from that of sub/super-threshold regime [9,10]. Kumar et al. [11] and Saurabh et al. [12] reveal the relationship of nominal delay with temperature in low voltage, which show the temperature has important effects in the low voltage. Kaul et al. [2] show that delay variability has a 2X improvement due to temperature effects. Consequently, the effects of temperature play an important role in the delay and delay variation, and cannot be ignored.

A lot of works have studied on the effect of temperature on the delay variation. In [13,14], they present the model of delay variability and temperature in the subthreshold regime, which is inversely proportional to temperature. Kiamehr et al. [15] introduce the adjustment of temperature-aware voltage scaling considering the process, but from the view of current, subthreshold regime is adopted. For stacked gates, a method is proposed in [16], which shows the dependence of mean and variance on temperature based on lognormal (*LN*) distribution for a CMOS inverter and gates with transistor

stacks in the subthreshold regime. Also, for stacked transistors, they assume that each transistor is independent. However, due to the connection of the intermediate node, the stacked transistors are dependent, and no longer follow the same Gaussian distribution. Nevertheless, the relationship of delay variation with temperature under near-threshold supply voltage has not been revealed.

For near-threshold region, Balef et al. [17] use log-skew-normal (*LSN*) distribution to model the delay distribution across all voltage regions with its distribution parameters determined by fitting from the Monte Carlo (MC) samples. But it does not reveal the relationship with the process parameters and the environment parameters and takes numerous MC simulations. A transregional current and delay model was developed in [18] in closed-form for near-threshold. But *LN* distribution is introduced for the delay variation computing. Nevertheless, due to the difference in current expression, *LN* is no longer suitable for near-threshold, so the computation of maximum/minimum delay is unsuitable. Therefore, it does not provide a reasonable analytical delay variation model for the near-threshold regime and does not have analysis on stacked gates.

In this paper, a novel analytical delay variation model with temperature in the near-threshold is proposed, and also extended to gates with stacked transistors. The main contributions of the work are as followed.

- A novel analytical delay variation model with temperature is derived by *LSN* distribution with moment matching for the near-threshold regime.
- Due to the existence of the intermediate node, the two stacked transistors are no longer independent. So a multi-variate threshold voltage approximation approach of stacked transistors is proposed for the computation of delay variation for staked gates with temperature consideration, which shows remarkable accuracy advantage compared with previous works.

The rest of the paper is organized as follows. Section 2 introduces the related properties of *LSN* distribution. The delay distribution models for a single transistor and stacked transistors are addressed in Section 3. For stacked transistors, a multi-variate threshold voltage approximation approach is proposed. And the delay variation model is derived by *LSN* distribution with moment matching. Section 4 shows the experimental results that three metrics (delay variability, maximum and minimum delay) are compared with SPICE MC simulation and other previous works. Section 5 concludes the paper.

## 2. The Properties of *LSN* Distribution

Since this paper mainly analyzes and derives based on *LSN* distribution, some basic properties used are introduced in this section.

### 2.1. Properties of Skew Normal Distribution

If $X$ is a random variable and follows a skew-normal (*SN*) distribution, that is $X \sim SN\left(\varepsilon, \omega^2, \lambda\right)$, its PDF ($f_{SN}(X)$) and CDF ($F_{SN}(X)$) are represented as

$$f_{SN}(X) = \frac{2}{\omega} \phi\left(\frac{X - \varepsilon}{\omega}\right) \Phi\left(\lambda \frac{X - \varepsilon}{\omega}\right) \tag{1}$$

$$F_{SN}(X) = \Phi\left(\frac{X - \varepsilon}{\omega}\right) - 2T\left(\frac{X - \varepsilon}{\omega}, \lambda\right) \tag{2}$$

where $\phi$ and $\Phi$ are the PDF and CDF of the standard normal distribution, $\varepsilon$, $\omega$, and $\lambda$ are the location, scale, and shape parameters of the *SN* distribution, $T(H, A)$ is Owen's *T* function and can be given by

$$T(H, A) = \frac{1}{2\pi} \int_0^A \frac{e^{\frac{-H^2}{2}(1+x^2)}}{1+x^2} dx \tag{3}$$

The mean $(\mu)$, variance $(\sigma^2)$, and skewness $(\gamma_1)$ (they are also called first, second and third moment) of the random variable $X$ are written in Equation (4) [19].

$$\begin{cases} \mu = \varepsilon + \omega\beta\sqrt{\frac{2}{\pi}} \\ \sigma^2 = \omega^2\left(1 - \frac{2}{\pi}\beta^2\right) \\ \gamma_1 = \frac{4-\pi}{2}\frac{\left(\frac{2}{\pi}\beta^2\right)^{\frac{3}{2}}}{\left(1 - \frac{2}{\pi}\beta^2\right)^{\frac{3}{2}}} \end{cases} \tag{4}$$

where

$$\beta = \frac{\lambda}{\sqrt{1+\lambda^2}} \tag{5}$$

*2.2. Properties of Log-Skew-Normal Distribution*

If $Y$ is a random variable and has an exponent relationship with $X$ $(Y = e^X)$, it follows log-skew-normal distribution, that is $Y \sim LSN\left(\varepsilon, \omega^2, \lambda\right)$. And its PDF $(f_{SN}(X))$ and CDF $(F_{SN}(X))$ are represented as

$$f_{LSN}(Y) = \frac{2}{\omega y}\phi\left(\frac{\ln(Y) - \varepsilon}{\omega}\right)\Phi\left(\lambda\frac{\ln(Y) - \varepsilon}{\omega}\right) \tag{6}$$

$$F_{LSN}(Y) = \Phi\left(\frac{\ln(Y) - \varepsilon}{\omega}\right) - 2T\left(\frac{\ln(Y) - \varepsilon}{\omega}, \lambda\right) \tag{7}$$

According to [19], the mean and variance of $Y$ are illustrated in Equation (8). Besides, the variability $(\sigma_Y/\mu_Y)$ of $Y$ can be easily computed by dividing $\sigma_Y$ by $\mu_Y$.

$$\begin{cases} \mu_Y = 2e^\varepsilon e^{\omega^2/2}\phi\left(\beta\omega\right) \\ \sigma_Y^2 = 2e^{2\varepsilon}e^{\omega^2}\left(e^{\omega^2}\phi\left(2\beta\omega\right) - 2\phi^2\left(2\beta\omega\right)\right) \end{cases} \tag{8}$$

For this work, maximum/minimum delay ($\pm3\sigma$ percentile points) is also the important metric. In order to obtain $\pm3\sigma$ percentile points, Equation (9) must be solved.

$$F_{LSN}(Y) = \Phi\left(\pm3\right) \tag{9}$$

But it cannot be easily solved for the $LSN$ distribution due to the existence of Owen's $T$ function. To resolve this issue, a reasonable approximation is introduced in this paper to transform the form of the Owen's $T$ function. If the value of shape parameter $(\lambda)$ in $LSN$ distribution is close to 1, Owen's $T$ function has the following function.

$$T(h, 1) = \frac{1}{2}\Phi(h)(1 - \Phi(h)) \tag{10}$$

Equation (9) can be derived by

$$\Phi^2\left(\frac{\ln(Y) - \varepsilon}{\omega}\right) = \Phi\left(\pm3\right) \tag{11}$$

It can be clearly shown from Equation (11) that the maximum/minimum delay at $3\sigma / -3\sigma$ percentile point of $Y$ locates at $Y_{max}/Y_{min}$, which is

$$\begin{cases} Y_{max} = e^{\varepsilon + \Phi^{-1}\left(\sqrt{\Phi(3)}\right)\omega} \approx e^{\varepsilon + 3.21\omega} \\ Y_{min} = e^{\varepsilon + \Phi^{-1}\left(\sqrt{\Phi(-3)}\right)\omega} \approx e^{\varepsilon - 1.79\omega} \end{cases} \tag{12}$$

According to the previous analysis, the variability and the maximum/minimum value of $Y$ can be obtained.

### 2.3. Conclusion of Different Distribution

This section will list the properties of three distributions, which are lognormal, log-skew-normal and Gaussian distribution and usually adopted in the path delay analysis. For skew-normal distribution, $\lambda$ controls the shape of the CDF, it is evident that when $\lambda = 0$, SN reduces to normal distribution with mean $\varepsilon$ and variance $\omega$. The characteristics of different distributions is summarized in Table 1.

**Table 1.** The property of lognormal, log-skew-normal and Gaussian distributions.

| | Lognormal | Log-Skew-Normal | Gaussian |
|---|---|---|---|
| X | $X \sim N\left(\varepsilon, \omega^2\right)$ | $X \sim SN\left(\varepsilon, \omega^2, \lambda\right)$ | $X \sim N\left(\varepsilon, \omega^2\right)$ |
| Y | $Y = e^X \sim LN\left(\varepsilon, \omega^2\right)$ | $Y = e^X \sim LSN\left(\varepsilon, \omega^2, \lambda\right)$ | $Y = X \sim N\left(\varepsilon, \omega^2\right)$ |
| $\mu$ | $e^{\varepsilon + \frac{\omega^2}{2}}$ | $2e^{\varepsilon}e^{\omega^2/2}\phi\left(\beta\omega\right)$ | $\varepsilon$ |
| $\sigma^2$ | $\left(e^{\omega^2}-1\right)\cdot\left(e^{\varepsilon+\frac{\omega^2}{2}}\right)^2$ | $2e^{2\varepsilon}e^{\omega^2}\left(e^{\omega^2}\phi\left(2\beta\omega\right)-2\phi^2\left(2\beta\omega\right)\right)$ | $\omega^2$ |
| $\frac{\sigma}{\mu}$ | $\sqrt{e^{\omega^2}-1}$ | $\frac{\sqrt{2e^{2\varepsilon}e^{\omega^2}\left(e^{\omega^2}\phi(2\beta\omega)-2\phi^2(2\beta\omega)\right)}}{2e^{\varepsilon}e^{\omega^2/2}\phi(\beta\omega)}$ | $\frac{\omega}{\varepsilon}$ |
| $Y_{max}$ | $e^{\varepsilon+3\omega}$ | $e^{\varepsilon+3.21\omega}$ | $\varepsilon+3\omega$ |
| $Y_{min}$ | $e^{\varepsilon-3\omega}$ | $e^{\varepsilon-1.79\omega}$ | $\varepsilon-3\omega$ |

## 3. Delay Variation Model Based on *LSN* Distribution

In this section, delay variation models of a single transistor and stacked transistors based on *LSN* distribution are introduced, respectively. By moment matching, the distribution parameters of *LSN* can be computed. The detailed process of the proposed model is introduced in the following.

### 3.1. Delay Variation Model for Single Transistor

Recently, Keller et al. [18] pointed out that near-threshold on-state current is expressed as follows.

$$I_{on} = \mu C_{ox}\left(n-1\right)\frac{W}{L}\left(\frac{kT}{q}\right)^2 K_0 e^{K_1 \frac{V_{DD}-[V_{th0}-\kappa(T-T_0)]}{nkT/q}+K_2\left(\frac{V_{DD}-[V_{th0}-\kappa(T-T_0)]}{nkT/q}\right)^2} \tag{13}$$

where $\mu$ is the mobility of carriers, $C_{ox}$ is the oxide capacitance for unit area, $n$ is the subthreshold slope factor, $W$ and $L$ are width and length of a transistor, $k$ is the Boltzmann constant, $q$ is the elementary charge, $K_0$, $K_1$ and $K_2$ are process-independent constants, which are 0.54, 0.69 and $-0.033$ [18]. $V_{DD}$ is the operation voltage, $V_{thn0}$ is the threshold voltage at the temperature $T_0$. $\kappa$ is the temperature coefficient for threshold voltage, and $T$ is the absolute temperature.

Simplicity, Equation (13) can be rewritten as

$$I_{on} = K_3 T^2 e^X \tag{14}$$

where $K_3$ is expressed as $\mu C_{ox}\left(n-1\right)\left(W/L\right)\left(k/q\right)^2 K_0$ .

Although $\mu$ is also related to temperature, the mobility contributes only 2% compared with thermal and threshold voltage [12]. Therefore, in this work, the dependence of mobility with temperature is also ignored.

And $X$ is noted as

$$X = K_1 \frac{V_{DD} - [V_{th0} - \kappa\,(T - T_0)]}{nkT/q} + K_2 \left( \frac{V_{DD} - [V_{th0} - \kappa\,(T - T_0)]}{nkT/q} \right)^2 \tag{15}$$

Since the random variable $X$ is a quadratic function of threshold voltage $V_{th}$ which follows Gaussian distribution, the mean ($\mu$), variance ($\sigma^2$), and skewness ($\gamma_1$) of $X$ can also be calculated by definition and expressed as

$$
\begin{cases}
\mu_X = \frac{qK_1}{nk}\frac{1}{T}V_{DD} - \frac{qK_1}{nk}\frac{1}{T}[\mu_0 - \kappa\,(T - T_0)] + \frac{q^2K_2}{(nk)^2}\frac{1}{T^2}V_{DD}{}^2 \\
\quad - 2\frac{q^2K_2}{(nk)^2}\frac{1}{T^2}V_{DD}[\mu_0 - \kappa\,(T - T_0)] + \frac{q^2K_2}{(nk)^2}\frac{1}{T^2}\left([\mu_0 - \kappa\,(T - T_0)]^2 + \sigma_0{}^2\right) \\
\sigma_X{}^2 = \frac{q^2K_2}{(nk)^2}\frac{1}{T^2}\left[2\sigma_0^4 + 4\sigma_0^2\left(V_{DD} - [\mu_0 - \kappa\,(T - T_0)] + \frac{nkK_1}{2K_2}T\right)^2\right] \\
\gamma_{1X} = \left(\begin{aligned}
& \left(\frac{q^2K_2}{(nk)^2}\frac{1}{T^2}\right)^3 E\left[V_{DT'}^6\right] \\
& -\frac{3}{4}\left(\frac{qK_1}{nk}\frac{1}{T}\right)^2\frac{q^2K_2}{(nk)^2}\frac{1}{T^2}E\left[V_{DT'}^4\right] \\
& +\frac{3}{16}\frac{q^2}{(nk)^2}\frac{K_1^4}{K_2}\frac{1}{T^2}E\left[V_{DT'}^2\right] - \frac{K_1^6}{64K_2^3}
\end{aligned}\right) \bigg/ \sigma_X{}^3 + \frac{-3\mu_X\sigma_X{}^2 - \mu_X{}^3}{\sigma_X{}^3}
\end{cases}
\tag{16}
$$

where $\mu_0$ and $\sigma_0$ are the mean and standard deviation of $V_{th0}$, and $E\left[V_{DT'}^6\right]$, $E\left[V_{DT'}^6\right]$, $E\left[V_{DT'}^6\right]$ can be computed by integration and the final form is illustrated in

$$
\begin{cases}
E\left[V_{DT'}^6\right] = 15\sigma_0^6 + 45\sigma_0^4(V_{DD}')^2 + 15\sigma_0^2(V_{DD}')^4 + (V_{DD}')^6 \\
E\left[V_{DT'}^4\right] = 3\sigma_0^4 + 6\sigma_0^2(V_{DD}')^2 + (V_{DD}')^4 \\
E\left[V_{DT'}^2\right] = \sigma_0^2 + (V_{DD}')^2
\end{cases}
\tag{17}
$$

where

$$V_{DD}' = \left(V_{DD} - [\mu_0 - \kappa\,(T - T_0)] + \frac{nk}{2q}\frac{K_1}{K_2}T\right)^2 \tag{18}$$

With a simper linear RC-delay model, the delay of a gate can be written as

$$T_d = k_f \frac{V_{DD}C_L}{I_{on}} = \frac{k_f V_{DD}C_L}{K_3}\frac{1}{T^2}e^{-X} \tag{19}$$

### 3.2. Delay Variation Model for Stacked Transistors

In this section, we broaden the proposed model to gates with stacked transistors , including stacked transistors, such as NAND, shown in Figure 1. Due to the existing of intermate node $V_x$, the stacked transistors are not independent. In order to extend the proposed model, a multi-variate threshold threshold voltage approximation approach is proposed.

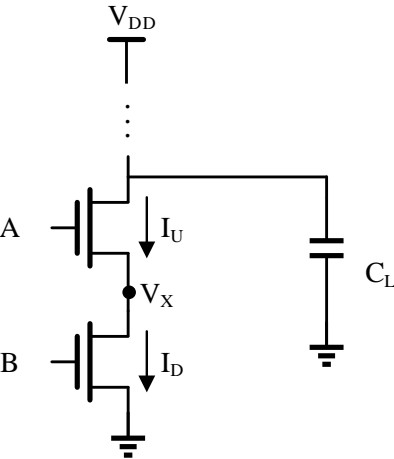

**Figure 1.** The schematic of stacked transistors.

$I_U$ and $I_D$ are the current through the up and down series-connected transistor as shown in Figure 1, which can be computed by

$$\begin{cases} I_U = K_3 T^2 e^{K_1 \frac{V_{DD}-V_X-V_{thU}}{nkT/q} + K_2 \left( \frac{V_{DD}-V_X-V_{thU}}{nkT/q} \right)^2} \cdot \left( 1 - e^{-\frac{V_{DD}-V_X}{kT/q}} \right) \\ I_D = K_3 T^2 e^{K_1 \frac{V_{DD}-V_{thD}}{nkT/q} + K_2 \left( \frac{V_{DD}-V_{thD}}{nkT/q} \right)^2} \cdot \left( 1 - e^{-\frac{V_X}{kT/q}} \right) \end{cases} \tag{20}$$

where $V_x$ is the intermediate node voltage of the two stacked transistors, $V_{thU}$ and $V_{thD}$ are the threshold voltage for up and down transistors and illustrated in Equation (21), respectively.

$$\begin{cases} V_{thU} = V_{thU0} + \kappa \left( T - T0 \right) \\ V_{thD} = V_{thD0} + \kappa \left( T - T0 \right) \end{cases} \tag{21}$$

where $V_{thU0}$ and $V_{thD0}$ are the threshold voltage of up and down transistor at temperature $T0$.

Due to the two transistors are series-connected, so the $I_U = I_D$. However, $V_x$ cannot be solved analytically, because it causes a transcendental equation. Therefore, a linear approximation method is introduced to determine the relationship between $V_{thU}$ and $V_{thD}$. Through running MC simulation for a NAND2 cell at 0.55 V and 25 °C, related simulation results are plotted in Figure 2, which show that the voltage $V_x$ changes approximate linearly with $V_{thU}$ and $V_{thD}$. Moreover, the slope with different temperatures can be considered constant whose specified fitting parameters are listed in Table 2. The maximum error is 3.12% across the entire temperature. Thereby, the equation form of $V_x$ can be determined by a bivariate linear model, which is shown in

$$V_X = k_U V_{thU} + k_D V_{thD} + k_C \tag{22}$$

where $k_U$, $k_D$, and $k_C$ are the fitting parameters.

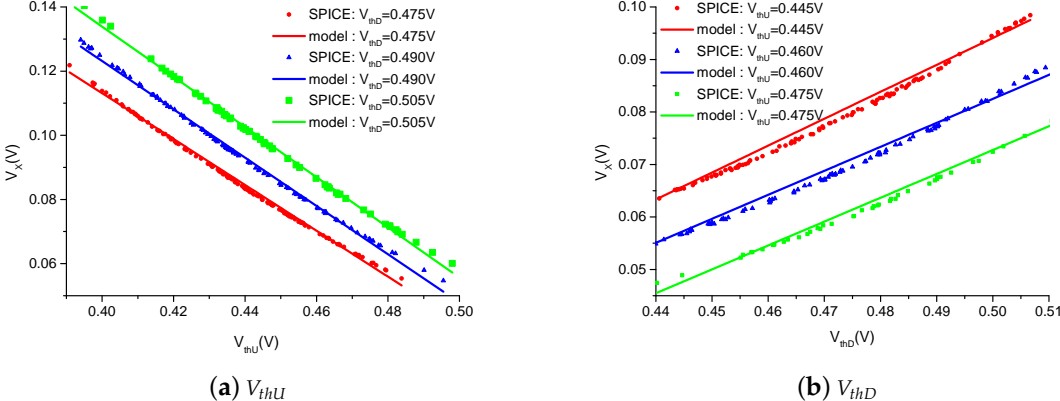

**(a)** $V_{thU}$　　　　　　　　　　　　　　　　　　　　**(b)** $V_{thD}$

**Figure 2.** The relationship of $V_x$ with $V_{thU}$ and $V_{thD}$: **(a)** $V_{thU}$; **(b)** $V_{thD}$.

**Table 2.** Coefficients and errors at different temperature.

| Temperature | $k_U$ | $k_D$ | $k_C$ | Error |
|---|---|---|---|---|
| $-25\,^\circ$C | $-0.70$ | 0.54 | 0.13 | 3.12% |
| 25 $^\circ$C | $-0.69$ | 0.53 | 0.14 | 2.64% |
| 75 $^\circ$C | $-0.68$ | 0.51 | 0.14 | 2.27% |
| 125 $^\circ$C | $-0.66$ | 0.50 | 0.15 | 1.98% |

By substituting $V_x$ into Equation (20), the discharge current for the NAND2 gate can be written as

$$I_{stack} = K_3 T^2 e^{K_1 \frac{V_{DD} - V_{th\_st}}{nkT/q} + K_2 \left( \frac{V_{DD} - V_{th\_st}}{nkT/q} \right)^2} \tag{23}$$

where $V_{th\_st}$ is the equivalent threshold voltage in the stacked gate and is given by

$$V_{th\_st} = (k_U + 1) \cdot (V_{thU0} + \kappa (T - T0)) + k_D (V_{thD0} + \kappa (T - T0)) + k_{C0} \tag{24}$$

Because the $V_{thU0}$ and $V_{thD0}$ both follow Gaussian distribution, the parameter of $V_{th\_st}$ can be obtained easily by Gaussian distribution operation.

Assuming that $V_{thU0} \sim N(\mu_{U0}, \sigma_{U0})$ and $V_{thD0} \sim N(\mu_{D0}, \sigma_{D0})$, the mean and variance of $V_{th\_st}$ can be expressed by

$$\begin{cases} \mu_{st} = (k_U + 1) \cdot (\mu_{U0} + \kappa (T - T0)) \\ \qquad + k_D (\mu_{D0} + \kappa (T - T0)) + k_{C0} \\ \sigma_{st} = \sqrt{(k_U + 1)^2 \sigma_{U0}^2 + k_L^2 \sigma_{L0}^2} \end{cases} \tag{25}$$

If X is noted as

$$X = K_1 \frac{V_{DD} - V_{th\_st}}{nkT/q} + K_2 \left( \frac{V_{DD} - V_{th\_st}}{nkT/q} \right)^2 \tag{26}$$

So we can expand the single transistor model to stacked transistors, and the expression of $T_{d\_st}$ is formulated by Equation (27), which has the same form as single transistor.

$$T_{d\_st} = \frac{k_f V_{DD} C_L}{K_3 T^2} e^{-X} \tag{27}$$

### 3.3. Moment Matching

According to [17], *LSN* distribution is suggested for the modeling of the delay distribution and shows remarkable accuracy in the near-threshold region. Therefore, *X* is assumed skew-normal distribution, the distribution parameters are $\varepsilon$, $\omega$ and $\lambda$, and the basic propertied of *SN* and *LSN* are introduced in Section 2. The specified flow of moment matching is illustrated in Figure 3. The mean, variance and skewness can be computed from *SN* distribution Equation (4) and definition Equation (16), respectively. Then, the three distribution parameters can be calculated by equaling the two equations, which are given as:

$$
\begin{cases}
\varepsilon = \mu - \sqrt{\frac{2}{\pi}} \omega \beta \\
\omega = \sqrt{\frac{\sigma^2}{1 - \frac{2}{\pi}\beta^2}} \\
\lambda = \sqrt{\frac{K^*}{1 + (\frac{2}{\pi} - 1)K^*}}
\end{cases}
\tag{28}
$$

where

$$
K^* = \frac{\pi}{2}\left(\frac{2}{4-\pi}\gamma_1\right)^{\frac{2}{3}}
\tag{29}
$$

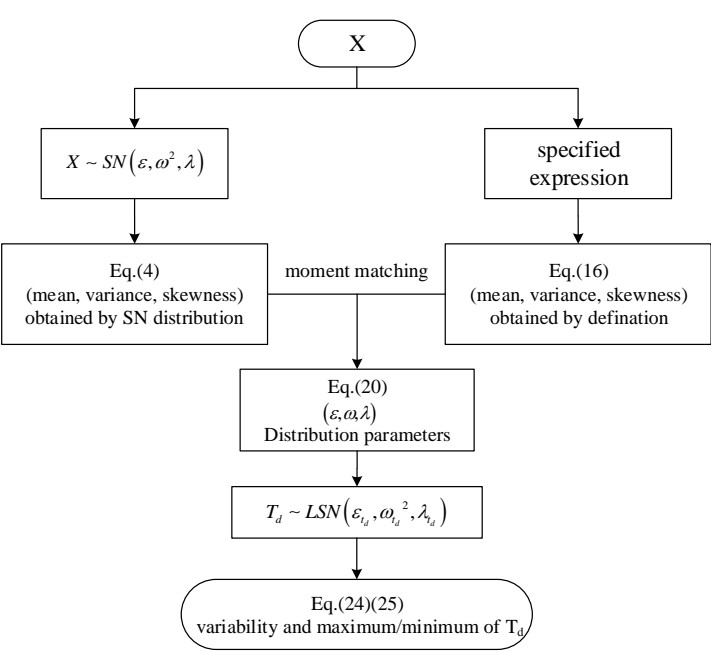

**Figure 3.** Flowchart of moment matching of log-skew-normal (*LSN*).

Equations (19) and (27) can be transformed into

$$
T_d = e^{-X + \ln\left(\frac{k_f V_{DD} C_L}{K_3}\frac{1}{T^2}\right)}
\tag{30}
$$

Thereby, three distribution parameters ($\varepsilon_{T_d}$, $\omega_{T_d}$ and $\lambda_{T_d}$) of $T_d$ are expressed as

$$
\begin{cases}
\varepsilon_{T_d} = -\mu_X + \ln\left(\frac{k_f V_{DD} C_L}{K_3} \frac{1}{T^2}\right) - \sqrt{\frac{2}{\pi}} \omega_{T_d} \beta_{T_d} \\[2mm]
\omega_{T_d} = \sqrt{\dfrac{\sigma_X^2}{1 - \frac{2}{\pi}\beta_{T_d}^2}} \\[2mm]
\lambda_{T_d} = \sqrt{\dfrac{K^*}{1 + \left(\frac{2}{\pi} - 1\right)K^*}}
\end{cases}
\tag{31}
$$

where $\beta_{Td} = \sqrt{\frac{K^*}{1 + \frac{2}{\pi}K^*}}$.

Then, according to Table 1, the variability and maximum/minimum of delay can be calculated.

$$
\left(\frac{\sigma}{\mu}\right)_{Td} = \frac{\sqrt{2 e^{2\varepsilon_{Td}} e^{\omega_{Td}^2}\left(e^{\omega^2}\phi\left(2\beta_{Td}\omega_{Td}\right) - 2\phi^2\left(2\beta_{Td}\omega_{Td}\right)\right)}}{2 e^{\varepsilon_{Td}} e^{\omega_{Td}^2/2}\phi\left(\beta_{Td}\omega_{Td}\right)}
\tag{32}
$$

$$
\begin{cases}
T_{d_{\max}} \approx e^{\varepsilon_{Td} + 3.21\omega_{Td}} \\
T_{d_{\min}} \approx e^{\varepsilon_{Td} - 1.79\omega_{Td}}
\end{cases}
\tag{33}
$$

The delay variability and maximum/minimum delay value increase with decreasing temperature and voltage. However, due to the near-threshold regime with complicated current expression, a log-skew-normal distribution is adopted and its distribution parameters are obtained by moment matching, they have no explicit relationship with temperature and voltage. But these expressions can be verified by the following experiments and described in the next section.

## 4. Experiments and Result Comparisons

According to the above proposed delay variation model, experiments are carried out on the gates with a single transistor and stacked transistors under all kinds of metrics, such as delay variability, and maximum/minimum delay value, in SMIC40nmLL technology. For a single transistor, we take INV as an example; for stacked transistors, NAND2 is used as an instance. Because the proposed method is analytical, it has the inherent advantage in terms of runtime. Taking NAND as an example, under the same server resources, it takes 40 s to obtain three metrics by SPICE MC method, while it only needs 0.6 s by the proposed model in this paper. Therefore, it has increased by more than 60 times in runtime. For the perspective of accuracy, detailed comparisons with the similar analytical methods in [13,16,18] are done. Since they do not have the contrast of $\pm 3\sigma$ point, the corresponding calculations are listed in Table 3 to allow a fair comparison with our work to validate the effectiveness of our model.

In order to validate our proposed model based on *LSN*, SPICE MC simulations are conducted to compare variability of INV and NAND2 with different temperatures at the supply voltage of 0.35 V and 0.55 V, respectively, as shown in Figure 4. The x-axis represents the temperature (from $-25\sim125\ ^\circ$C) and the y-axis represents the value of current variability (left figures) or the relative error (right figures). Besides, different combinations of gates and voltages are illustrated in Figure 4a–d, respectively. The four left figures show that the delay variabilities are greater in 0.35 V than that of in 0.55 V, and have a tendency that they increase with decreasing temperature in all model. In all models, our *LSN* model and MC simulation match best with maximum error 5% in NAND. For INV, due to the same current formula with [18], the delay variabilities are the same as each other, which are better than the result of *LN* distribution in [13,16]. Besides, in terms of the complex gate at 0.55 V, the stacked threshold approximation is more suitable and the error is about 2% at the 0.55 V which is less than 10% and 20% in the others. It improves 5X at least in accuracy. In addition, the error keeps constant across the entire temperature, which shows the excellent stability.

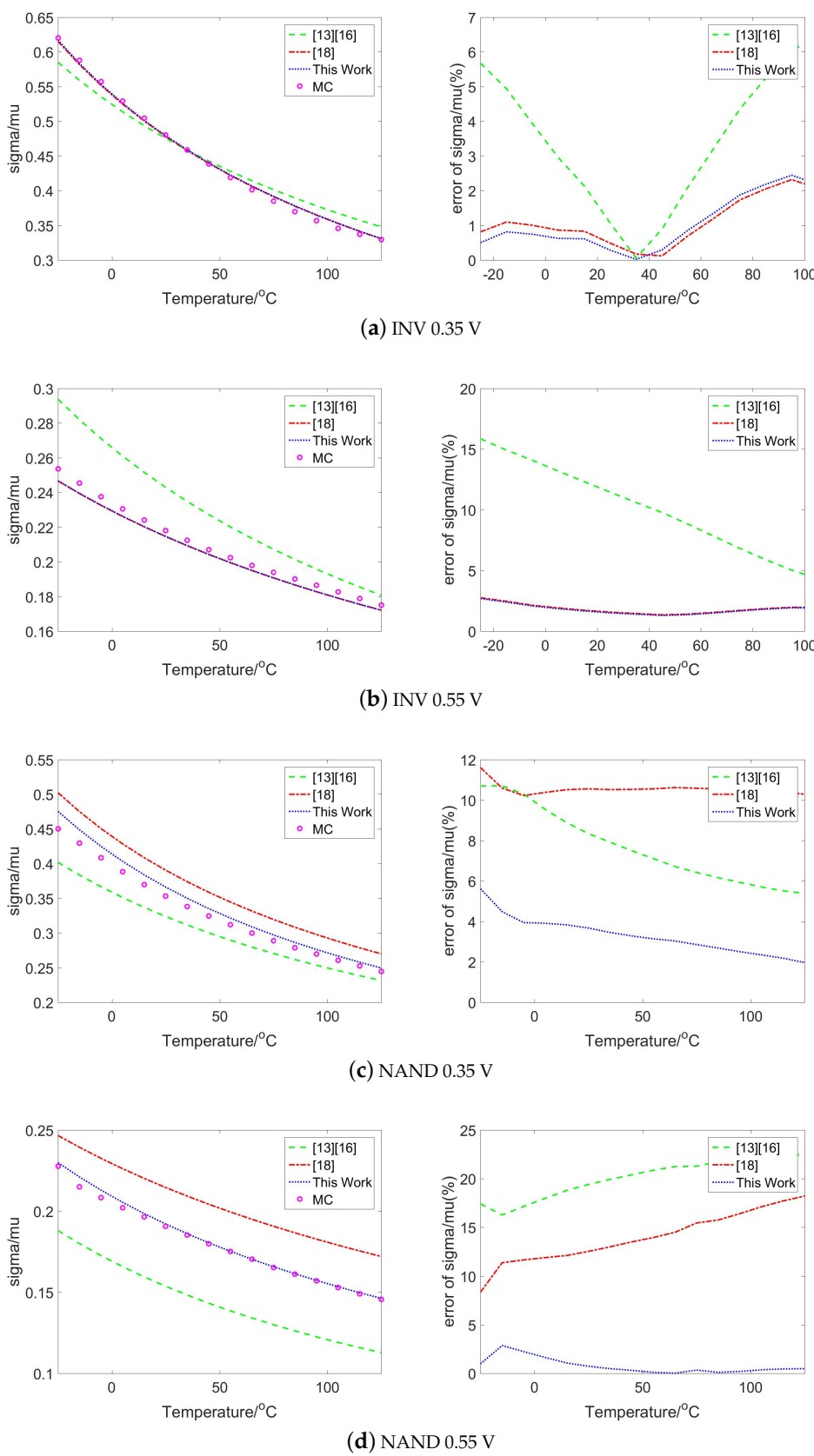

**Figure 4.** Comparison of delay variability with different temperature under different gates and voltages: (**a**) INV 0.35 V; (**b**) INV 0.55 V; (**c**) NAND 0.35 V; (**d**) NAND 0.55 V.

**Table 3.** Comparison of methods in different works.

| | [13,16] | [18] | This Work |
|---|---|---|---|
| Distribution | *LN* | *LN* | *LSN* |
| Current | $I_{ON} = I_0 \cdot e^{\frac{q}{nkT}(V_{DD}-V_{th})}$ | $I_{on} = I_0 K_0 e^{K_1 \frac{(V_{DD}-V_{th})}{nkT/q} + K_2 \left(\frac{V_{DD}-V_{th}}{nkT/q}\right)^2}$ | |
| Stack threshold | $V_{th\_st} = V_{thU} = V_{thD}$ | | $V_{th\_st} = f(V_X)$ |
| Delay variability | Equations (15) and (5) in [13,16] | Equation (58) | Equation (24) |
| $+3\sigma$ | $e^{\varepsilon+3\omega}$ | | Equation (25) |
| $-3\sigma$ | $e^{\varepsilon-3\omega}$ | | Equation (25) |

The specified Equations (15) and (5) in [13,16] is given in $\frac{\sigma}{\mu} = \sqrt{\exp\left[\left(\frac{q\sigma_{V_{th}}}{nkT}\right)^2\right] - 1}$. The specified Equation

(58) in [18] is given in $\frac{\sigma}{\mu} = \dfrac{\sqrt{\int_{-\infty}^{\infty} \frac{C_{load}^2}{I_F^2} \frac{V_{DD}^2}{\sigma_{V_t}\sqrt{2\pi}} e^{-2k_1 \frac{V_{DD}-V_t}{nkT/q} - 2k_2\left(\frac{V_{DD}-V_t}{nkT/q}\right)^2 - \frac{(V_t-uV_t)^2}{2\sigma_{V_t}^2}} dV_t - \left(E[t_{pd}(V_t)]\right)^2}}{\int_{-\infty}^{\infty} \frac{C_{load}}{I_F} \frac{V_{DD}}{\sigma_{V_t}\sqrt{2\pi}} e^{-k_1 \frac{V_{DD}-V_t}{nkT/q} - k_2\left(\frac{V_{DD}-V_t}{nkT/q}\right)^2 - \frac{(V_t-uV_t)^2}{2\sigma_{V_t}^2}} dV_t}$.

Figure 5 is the dependence of maximum delay on temperature, which still presents the relationship of INV and NAND at different voltage across the entire temperature range. The four figures show that the maximum delays increase with decreasing voltage and temperature, due to lower current at the low voltage and low temperature. From the left absolute value figures of Figure 5b,d, the *LSN* model is agreement with MC simulation; and the right error figures show that the errors of *LSN* model are less than 5% and keep stable across the entire temperature range. Figure 6a,c show the dependence with temperature in 0.35 V. Method in [18] performs best for INV and worst for NAND, but [13,16] perform worst for INV and have the same accuracy with proposed method for NAND. Therefore, from overall view, *LSN* performs better in the two kinds of gates compared with the published works.

Figure 6 shows the result of minimum delay. The four figures show that the minimum delays have same tendency with maximum, which are increase with decreasing voltage and temperature. From left figures, no matter under what kinds of gates and voltage, the *LSN* model and MC data show remarkable consistency with each other. More detailed error information can be seen from error figures. The error is almost 2% in all four conditions, which improves 5X–10X and shows a distinct advantage over the other two state-of-the-art works. Therefore, as for $-3\sigma$, the proposed model features more obvious advantages, because of the reasonable of *LSN* equivalent model and the effectiveness of the threshold approximation of stacked transistors.

Above all, *LSN* model proposed in this work outperforms the model in previous works under different voltages and gates across entire temperature and has a strong agreement with SPICE MC simulation result.

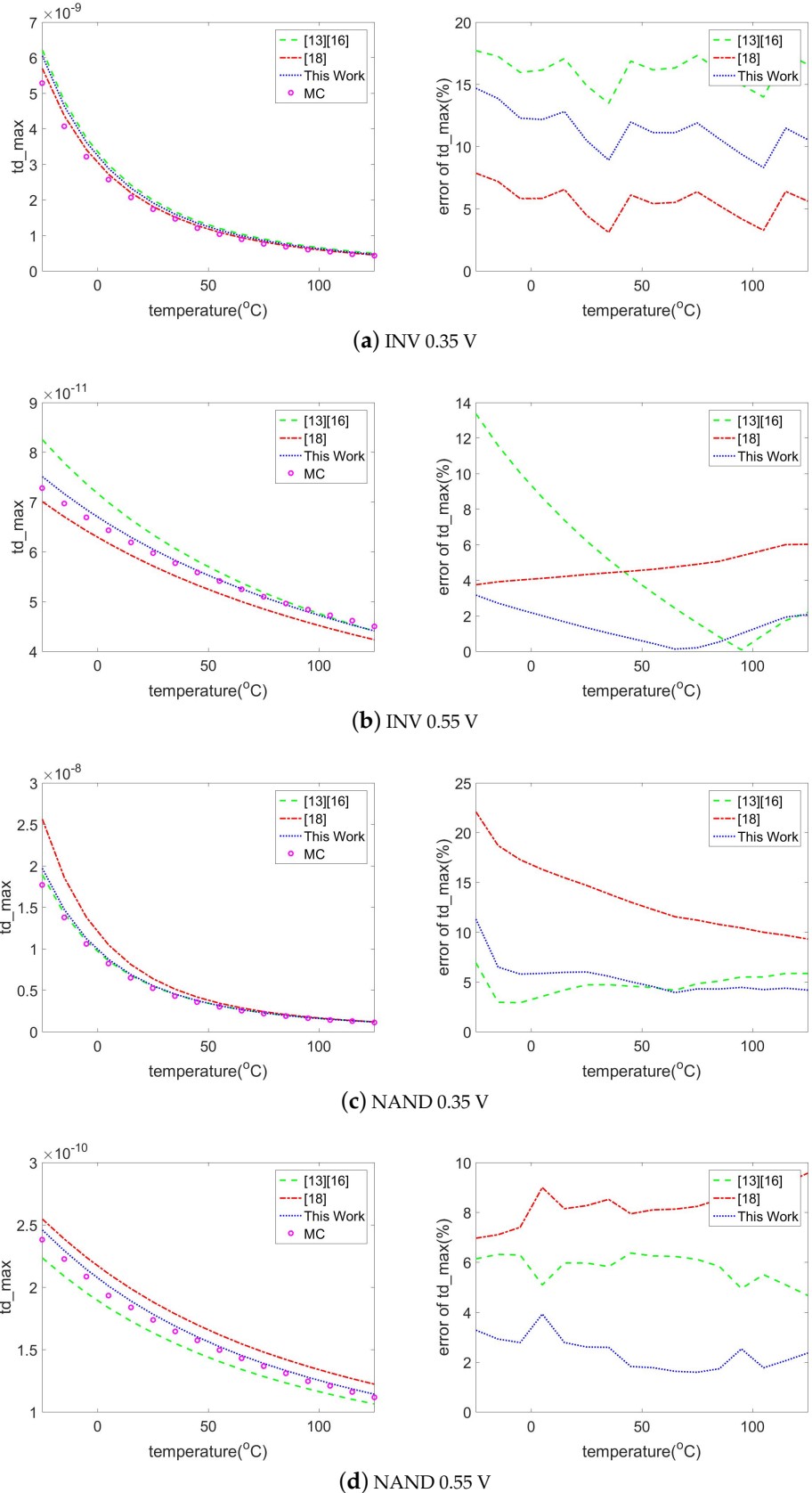

**Figure 5.** Comparison of maximum delay with different temperature under different gates and voltages:
(**a**) INV 0.35 V; (**b**) INV 0.55 V; (**c**) NAND 0.35 V; (**d**) NAND 0.55 V.

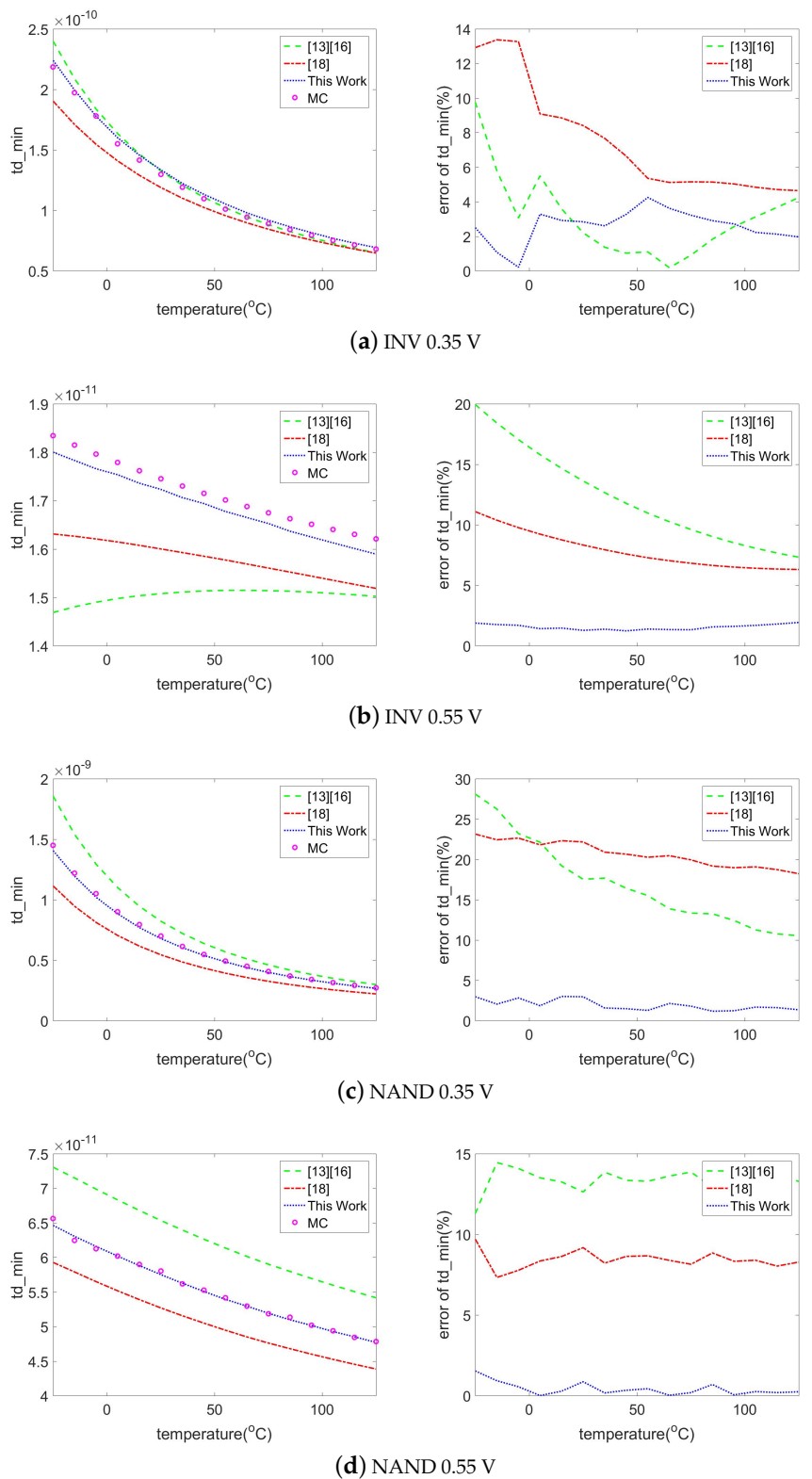

**Figure 6.** Comparison of minimum delay with different temperature under different gates and voltages: (**a**) INV 0.35 V; (**b**) INV 0.55 V; (**c**) NAND 0.35 V; (**d**) NAND 0.55 V.

## 5. Conclusions

This paper proposes a *LSN*-based methods for current and delay variation model. In order to obtain the distribution parameters, the moment matching is adopted. Besides, a multi-variate threshold

voltage approximation approach of stacked transistors is proposed and makes the method easily extend to stacked gate. The model is validated at different voltages and gates across the entire temperature. In addition, it also provides a detailed analysis on the dependence of the three important metrics (delay variability and maximum/minimum delay value)on the temperature, which are consistence to SPICE MC simulation with maximum delay variability 5% and performs very well , especially at minimum delay with a 5X–10X error improvement compared with other works.

**Author Contributions:** J.G., P.C. and J.Y. organized this work. J.G., J.W. and Z.L. performed the modeling, simulation and experiment work. The manuscript was written by J.G. and P.C., and edited by J.G.

**Funding:** This research was funded Supported by the Open Project Program of the State Key Laboratory of Mathematical Engineering and Advanced Computing, the National Science and Technology Major Project (Grant No. 2017ZX01030101), and National Natural Science Foundation of China (Grant No. 61834002).

**Acknowledgments:** The authors thank Hao Yan for his helpful insight and suggestions.

**Conflicts of Interest:** The authors declare no conflict of interest.

## Abbreviations

The following abbreviations are used in this manuscript:

SN    skew-normal
LSN   log-skew-normal
MC    Monte Carlo

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
