# Peer review of "Analytical Gate Delay Variation Model with Temperature Effects in Near-Threshold Region Based on Log-Skew-Normal Distribution"

_electronics, doi:10.3390/electronics8050501_

Round 1
Reviewer 1 Report
1) Language must be improved
2) The Error plots can be made into separate figures, smaller figures are difficult to read
3) Some explanation and physical insight of the final derived equations would be useful.
4) Eq 19, the final delay equation says the delay of a gate is proportional to VDD & CL, and not VDD^2?
5) Would use a different variable name for in equation 14 as C is confusing with capacitor
Author Response
Dear reviewer,
Thank you for you help and comments.
The point-to-point response is in the attachment.
Best wishes !

Reviewer 2 Report
Reviewer thinks the paper is interesting for some researchers. However, he suggests the authors
should revise it for publication as follows.
1) Please care how to use LSN, LN, MC, and SN in Abstract and Introduction.
- l. 5
log-skew-normal (LSN) -> log-skew-normal
- l.9
MC -> Monte-Carlo
- l. 10
LN -> lognormal
- p. 1, l. 31
LN -> lognormal (LN)
- p.2, l. 40
lognormal (LN) -> LN
- p.2, l. 65
skew-normal -> skew-normal (SN)
2) "subthreshold" is adjective. So please revise in the following ways.
- l. 21
sub/super-threshold -> sub/super-threshold regime
- l. 27 and 29
subthreshold -> subthreshold regime
3) "spice" must be "SPICE"
4) After Eq.(2)
Please use the same font of (Greek) phi in the text as in Eqs.(1) and (2).
5) Please check some equations as follows.
- Eq.(8)
left hand side of Eq. on sigma_Y must have root.
- Eq. (11)
omega_Y -> omega
epsilon_Y -> epsilon
- sigma of LSN in Table 1
The same as in Eq.(8)
- Eq.(13)
+ + K_2 -> + K_2
- Eq.(27) must be in the same way as Eq.(19)
K_f -> k_f
T^2 -> C T^2
- p.9, before the 120th line
The numerator must have root, and "E[t_pd(V_t)]" must be squared.
Please check original equation.
6) Eq.(16)
Please describe what mu_0 and sigma_0 mean. Reviewer guesses
they are related to V_th0.
7) Please check English carefully. For example,
- p. 9, l. 118
coontrast -> contrast
- p.11, l. 144
model obviously features more obvious advantages
-> model deatures more obvious advantages
8) In References
- Ref.[10] : 84-114 -> 83-114
- Ref.[14] : "Journal Abbreviation" must be revised.
9) p. 5, l. 92
Figure 2 -> Figure 1
10) Body bias is sometimes useful as shown in the following paers.
- M. Sumita, S. Sakiyama, M. Kinoshita, Y. Araki, Y. Ikeda, and K. Fukuoka,
"Mixed Body Bias Techniques With Fixed Vt and Ids Generation Circuits,"
IEEE J. Solid-State Circuits, vol. 40, no. 1, pp. 60-66, Jan. 2005.
- J. Wang, K. Yasue, T. Matsuoka, and K. Taniguchi,
"A Design for Ultra-Low-Voltage CMOS Digital Circuits with Performance Characteristics Compensation,"
Far East J. Electronics and Communications, Vol. 5, No. 1, pp. 59-65, Sep. 2010.
However, the paper does not describe how to deal the body of MOS devices.
It is more important in stacked device case.
Please comment it.
11) Abstract describes 2X improvement in stacked gates.
However, it is not so clear in Sec. 4. Please describe it more obviously in Sec. 4.
12) The authors use both kT/q and phi_t in some equations.
Please use one of them.
Author Response
Dear reviewer,
Thank you for your help and comments.
The point-to-point response is in the attachment.
Best Wishes !
